# Comparison of the Microbiome-Metabolome Response to Copper Sulfate and Copper Glycinate in Growing Pigs

**DOI:** 10.3390/ani13030345

**Published:** 2023-01-19

**Authors:** Hulong Lei, Qian Du, Naisheng Lu, Xueyuan Jiang, Mingzhou Li, Dong Xia, Keren Long

**Affiliations:** 1Key Laboratory of Livestock and Poultry Resources (Pig) Evaluation and Utilization of Ministry of Agriculture and Rural Affairs, Shanghai Engineering Research Center of Breeding Pig, Institute of Animal Husbandry & Veterinary Sciences, Shanghai Academy of Agricultural Sciences, Shanghai 201106, China; 2Institute of Animal Genetics and Breeding, College of Animal Science and Technology, Sichuan Agricultural University, Chengdu 611130, China

**Keywords:** copper sulfate, copper glycinate, microbiome, metabolome, pigs, copper source

## Abstract

**Simple Summary:**

Copper sulfate and copper glycinate have been used as additives for copper supplementation and growth promotion in the swine feed industry. However, their effects on the gut microenvironment, especially for the fecal microbiota and metabolites, still remain unclear, which are the important indicators of gut health. As well as the nutrient digestibility and physiological and biochemical indices of pigs, the effects of copper sulfate and copper glycinate on fecal microbial communities and metabolic profiles were investigated to understand it better in the present study. Our results suggested that copper sulfate and copper glycinate could differentially affect fecal microbiota and metabolites, by which dietary copper glycinate exerted more beneficial effects on the gut health of pigs.

**Abstract:**

This study aims to compare the fecal microbiome-metabolome response to copper sulfate (CuSO_4_) and copper glycinate (Cu-Gly) in pigs. Twelve Meishan gilts were allocated into the CuSO_4_ group and the Cu-Gly group (fed on a basal diet supplemented with 60 mg/kg copper from CuSO_4_ or Cu-Gly) paired in litter and body weight. After a two-week feeding trial, the Cu-Gly group had a higher copper digestibility, blood hemoglobin, and platelet volume and higher levels of plasma iron and insulin-like growth factor-1 than the CuSO_4_ group. The Cu-Gly treatment increased the abundance of the *Lachnospiraceae* family and the genera *Lachnospiraceae XPB1014*, *Corprococcus_3*, *Anaerorhabdus_furcosa_group*, *Lachnospiraceae_FCS020_group*, and *Lachnospiraceae_NK4B4_group* and decreased the abundance of the *Synergistetes* phylum and *Peptostreptococcaceae* family compared to the CuSO_4_ treatment. Moreover, the Cu-Gly group had a lower concentration of 20-Oxo-leukotriene E4 and higher concentrations of butyric acid, pentanoic acid, isopentanoic acid, coumarin, and Nb-p-Coumaroyl-tryptamine than the CuSO_4_ group. The abundance of *Synergistetes* was positively correlated with the fecal copper content and negatively correlated with the fecal butyric acid content. The abundance of the *Lachnospiraceae_XPB1014_group* genus was positively correlated with the plasma iron level and fecal contents of coumarin and butyric acid. In conclusion, Cu-Gly and CuSO_4_ could differentially affect fecal microbiota and metabolites, which partially contributes to the intestinal health of pigs in different manners.

## 1. Introduction

Copper (Cu) is essential for animal growth, erythropoiesis, and metabolisms [1]. Copper sulfate (CuSO_4_) has been used as an additive at high doses for reducing diarrhea frequency and improving the growth performance of piglets over the last decades [2,3]. However, Cu chelate with amino acid or peptide (organic Cu) is a potential substitute for CuSO_4_ in the feed industry. Previous studies suggested that the improvements of the Cu-lysine complex in the daily gain and feed intake of piglets were greater than those of CuSO_4_ [4], and 100 mg/kg Cu from Cu proteinate was as effective as 250 Cu mg/kg from CuSO_4_ in improving the growth performance of growing pigs [5]. Finishing pigs fed with 200 mg/kg Cu as Cu-lysine had a higher weight gain than those fed with equivalent Cu as CuSO_4_ [6]. Similarly, the growth performance of pigs fed with 50–150 mg/kg Cu as Cu methionine hydroxy analog chelated was higher than that of those fed with 160 mg/kg Cu as CuSO_4_ [7]. These suggest that organic Cu could improve the growth performance and Cu utilization in pigs compared to CuSO_4_, whereas the underlying mechanism remains unclear.

Gut microbiota co-evolve with the host in a symbiotic relationship for nutrient digestion [8] and energy metabolism and immunity [9]. However, the dietary composition affects the gut microenvironment and consequently influences the gut microbial composition, and metabolites are the key mediators of the interaction between gut microbiota and the host [10]. Previous investigations suggested that 300 mg/kg Cu from CuSO_4_ reduced the relative abundance of *Coprococcus* (*Lachnospiraceae* family) and modulated microbial metabolism pathways in the gut, which may further affect the health of suckling piglets [11]. Supplemental 200 mg/kg CuSO_4_ increased the richness of *Escherichia coli* in the hindgut and altered the gut microbial metabolisms of piglets, including energy metabolism, protein metabolism, and amino acid biosynthesis [12]. Furthermore, gut microbiota can transport, metabolize, and utilize the trace elements for survival through the assimilation from diets or competing with the host [13], but the host sequester traces elements and prevents bacterial access to them through the complicated mechanisms [14]. Compared to CuSO_4_, finishing pigs supplemented with Cu glycinate (Cu-Gly) exhibited increased Cu utilization and an abundance of *Clostridiaceae_1* in the colonic content [15]. Different from finishing pigs, the evolution of intestinal microbial communities is a more progressive [16] and variable procession in weanling and growing pigs [17]. Nevertheless, the effects of dietary Cu on the gut microbiota may depend on the experimental conditions, including the pig breed, copper source, and dosage [18]. Further studies on the mechanisms of the Cu source regulating the gut microbiota and mineral digestibility would help us to understand it better.

These findings suggested that CuSO_4_ and Cu-Gly may differentially affect the gut microbiome and metabolites, which partially at least play a role in the absorption of trace elements, growth axis hormones, and gut health. In order to verify the hypothesis, this study was conducted to investigate the effects of CuSO_4_ and Cu-Gly on the microbial communities, and metabolic profiles in the feces of growing pigs were investigated to understand them better in the present study; meanwhile, the nutrient digestibility and physiological and biochemical indices of pigs were determined.

## 2. Materials and Methods

### 2.1. Animals, Diets, and Experimental Design

Animal care and use procedures were followed in compliance with the Guidelines of the Animal Ethics Committee of Shanghai Academy of Agricultural Sciences, in accordance with standard international regulations. The experimental protocols used for research purposes in the present study were approved by the Institutional Review Board (SAASPZ0520011).

Twelve Meishan gilts (21.6 ± 1.33 kg) from three litters with the same parity were pre-treated with a basal corn-soybean diet (Table 1, without Cu addition, the intrinsic Cu level is 5.7 mg/kg) for two weeks. Afterward, the pigs were allocated to the CuSO_4_ group and the Cu-Gly group paired in litter and body weight (BW). Pigs in the CuSO_4_ group were fed the basal diet supplemented with 60 mg/kg Cu from Cu sulfate pentahydrate (CuSO_4_·5H_2_O, 25% Cu, Chelota Bio-tech Ltd., Sichuan, China), and the Cu-Gly-treated pigs were fed the basal diet supplemented with equivalent Cu from the Cu-Gly complex (21% Cu and 25% glycine, Debon Bio-tech Ltd., Hunan, China) for another two weeks, respectively. Then, 72.15 mg/kg glycine (99%, Aladdin, Shanghai, China) was added in the CuSO_4_-treated diet to achieve an equivalent glycine content to that in the Cu-Gly-treated diet. Pigs were individually raised in cages, receiving nipple water and feed ad libitum. 

The BW and feed consumption were recorded at 8 a.m. on d14; the average daily body weight gain (ADG) and average daily feed intake (ADFI) were determined. The feed efficiency was calculated by the feed intake to the body weight gain.

### 2.2. Sample Collections and Preparations

Feces samples were collected by rectal distention stimulation at the beginning (d0, baseline) and the end of the experiment (d14). Blood samples were collected into vacuum tubes by jugular vein puncture. Plasma was separated by centrifuging at 3000 rpm for 15 min and stored at −20 °C until the subsequent analysis.

### 2.3. Chemical Analysis of Diets and Feces

Feed and feces samples were dried at 105 °C until they reached a constant weight to determine the dry matter (DM) and milled through a 40 mesh screen. Crude protein (CP) was calculated through the determination of the nitrogen (N) content by the Kjeldahl method according to the Association of Official Analytical Chemists (AOAC, 2005). The concentrations of Cu, zinc (Zn), iron (Fe), and manganese (Mn) in diets and feces were determined using an atomic absorption spectrometer (novAA 350, Analytik Jena AG, Jena, Germany) (ISO 6869: 2000). The acid-insoluble ash (AIA) method was used to measure the apparent total tract digestibility (ATTD) of DM, CP, Cu, Zn, Fe, and Mn, as previously described [19].

### 2.4. Measurements of Blood Physiological and Biochemical Indices

The whole blood samples were immediately used for hematology analysis using an automatic hematology analyzer (Sinothinker Technology Co., Ltd., Shenzhen, China), including the numbers of white blood cells (WBC), lymphocytes (LYM), intermediate cells (MID), neutrophilic granulocytes (GRAN), red blood cells (RBC), and platelets (PLT), as well as the hemoglobin content (HGB), mean corpuscular hemoglobin (MCH), hematocrit (HCT), mean platelet volume (MPV), platelet distribution width (PDW), plateletcrit (PCT), and platelet–larger cell ratio (P–LCR).

To determine the plasma concentrations of trace elements, 1 mL plasma sample and 6 mL nitric acid were piped into a digestion tank and digested using a closed microwave digestion system (MARS6, CEM, Matthews, NC, USA) after standing still for 2 h of cold digestion. Concentrations of plasma Cu, Zn, Fe, and Mn were determined using inductively coupled plasma mass spectrometry (iCAP TQ ICP-MS, Thermo Scientific, Waltham, MA, USA).

According to the instructions of the manufacturer, the levels of growth hormone (GH) and insulin like growth factor 1 (IGF-1) in plasma were determined with the competitive ELISA kits (Elixir Canada Medicine Company Ltd., Vancouver, BC, Canada) for porcine GH (G066SC, intra-assay CV < 5.0%, inner-assay CV < 8.3%, sensitivity is 1.0 ng/mL) and IGF-1 (I060SC, intra-assay CV < 5.3%, inner-assay CV < 6.2%, sensitivity is 1.0 ng/mL), respectively. 

### 2.5. Pyrosequencing Analysis of Fecal Microbiome

The total genomic DNA of bacteria was extracted from the feces samples using the Ultra clean fecal DNA isolation kits (D1600, Solarbio Science & Technology, Beijing, China), according to the manufacturer’s protocol. The final DNA concentrations were determined by a Nano-Drop 1000 UV spectrophotometer (Thermo Scientific, Wilmington, DE, USA) at 260 and 280 nm, and the DNA quality was checked by 1% agarose gel electrophoresis. 

The V3-V4 hypervariable regions of the bacterial 16S rRNA gene were amplified for high-throughput sequencing by polymerase chain reactions (PCR) with the universal primers 338F (5′-ACTCCTACGGGAGGCAGCAG-3′) and 806R (5′-GGACTACHVGGGTWTCTAAT-3′). Purified amplicons were purified with the QIA quick PCR purification kits (28104, Qiagen, Valencia, CA, USA). The products were quantified using the QuantiFluorTM-ST fluorescent quantitation system (Promega, Madison, WI, USA), pooled in equimolar ratios and sequenced on an Illumina MiSeq PE250 platform (Illumina, San Diego, CA, USA) according to the standard protocols of Majorbio Bio-Pharm Technology Co. Ltd. (Shanghai, China). Alpha and beta diversity analyses were performed to describe the microbial diversity among samples using the online platform of the Majorbio Cloud Platform (www.majorbio.com, accessed on 20 December 2022). 

### 2.6. Liquid Chromatograph–Mass Spectrometer (LC–MS) Metabolomics Analysis

Fifty milligrams of feces samples were accurately weighed and extracted using a 400 µL methanol:ultrapure water (4:1) solution, with 0.02 mg/mL L-2-chlorophenylalanin as the internal standard. The mixture was settled at −10 °C, treated using a high-throughput tissue crusher (Wonbio-96c, Wanbo biotechnology Co., Ltd., Shanghai, China) at 50 Hz for 6 min, and then sonicated at 40 kHz for 30 min at 5 °C. The mixture was placed at −20 °C for 30 min to precipitate proteins and centrifuged at 13,000× *g* for 15 min at 4 °C. The supernatants were collected and carefully transferred to sample vials for the LC-MS analysis. 

The LC-MS analysis was performed on a Thermo ultra-high-performance liquid chromatography system (UHPLC) coupled with a Thermo Q Exactive mass spectrometer (MS). Separation was achieved with an Acquity BEH C18 column (100 mm × 2.1 mm, i.d. 1.7 μm, Waters, Milford, CT, USA). The mobile phases consisted of 0.1% formic acid in water:acetonitrile (95:5, solvent A) and 0.1% formic acid in acetonitrile:isopropanol:water (47.5:47.5:5, solvent B). The sample injection volume was 2 µL, and the flow rate was 0.4 mL/min. The column temperature was maintained at 40 °C. During the analysis period, samples were stored at 4 °C.

The raw data of the UHPLC-MS analysis were imported into Progenesis QI 2.3 (Nonlinear Dynamics, Waters Corporation, Milford, MA, USA) for peak detection and alignment. The preprocessing results generated a data matrix that consisted of the retention time, mass-to-charge ratio (m/z) values, and peak intensity. The internal standard was used for data quality control (QC), and metabolic features were discarded when the relative standard deviation of QC > 30%. The mass spectra of these metabolic features were identified by using the accurate masses, MS fragments spectra and isotope ratio difference with searching in reliable biochemical databases as Human Metabolome Database (HMDB) (http://www.hmdb.ca, accessed on 16 December 2022) and Metlin database (https://metlin.scripps.edu, accessed on 25 December 2022).

### 2.7. Determination of Short-Chain Fatty Acids (SCFAs)

Gas chromatography analysis was performed to determine the concentrations of SCFAs in feces, as previously described [20]. A total of 1 g of each feces sample was diluted with distilled water, homogenized, and centrifuged (Heraeus Instruments, Düsseldorf, Germany) at 11,900× *g* for 15 min. Subsequently, 0.5 μL supernatant was injected into the gas chromatograph (7890B, Agilent Technologies, Santa Clara, CA, USA) equipped with a capillary column (30 m × 0.32 mm × 0.25 mm film thickness; Varian, Inc., Santa Clara, CA, USA) after filtrating through a 0.22 mm filter. The column, injector, and detector temperatures were 130 °C, 180 °C, and 180 °C, respectively. Crotonic acid was used as an internal standard. A standard SCFAs mixture including acetic acid, propionic acid, butyric acid, pentanoic acid, and isopentanoic acid was used for calculation, and the results were expressed as the mg/g of a fresh feces sample.

### 2.8. Statistical Analysis

Data organization and scientific graphing were performed using Microsoft Excel (Redmond, WA, USA) and GraphPad Prism (Version 8.0; La Jolla, CA, USA). Data analysis was performed using IBM SPSS Statistics 20.0 (IBM SPSS Statistics, Chicago, IL, USA). The statistical significance of differences was analyzed by a t-test for the paired samples of the CuSO_4_ and Cu-Gly groups, as well as paired samples before (d0, baseline) or after dietary Cu supplementation (d14). After confirming the normal distribution on histograms, the correlations between the microbial composition and metabolites in feces were assessed by Pearson’s correlation test using R language and the Pheatmap package. The Spearman correlations between the microbial composition and the concentrations of fecal SCFAs, trace elements, or plasma indices were analyzed on the online platform of Majorbio Cloud Platform (www.majorbio.com, accessed on 20 December 2022). All the differences were considered as statistically significant at *p* < 0.05.

## 3. Results

### 3.1. Growth Performance, Digestibility, and Fecal Microelement Contents

As shown in Table 2, Cu-Gly supplementation significantly increased the ADFI of pigs (*p* = 0.039) compared with CuSO_4_ supplementation. There was no difference in the final BW, ADG, and feed efficiency between the CuSO_4_ and Cu-Gly treatments. Compared to the CuSO_4_ treatment, the Cu-Gly treatment improved the ATTD of dietary Cu (*p* = 0.048), Fe (*p* = 0.021), and Mn (*p* = 0.048) and decreased the fecal contents of Cu (*p* = 0.023), Fe (*p* = 0.016), and Mn (*p* = 0.040). However, no differences in the ATTD of dietary DM, CP, and Zn as well as in their contents in feces were observed between the CuSO_4_ and Cu-Gly treatments.

### 3.2. Hematology and Plasma Indicators

After two weeks of the experiment, the HGB (*p* < 0.001, *p* = 0.006. Figure 1a) and plasma Cu concentration (*p* = 0.049, 0.013. Figure 2a) increased in the CuSO_4_ and Cu-Gly treatments. Pigs supplemented with Cu-Gly had an increase in MPV (*p* = 0.001. Figure 1b), PDW (*p* < 0.001. Figure 1c), and P-LCR (*p* = 0.041. Figure 1d) and a higher plasma Fe concentration (*p* = 0.001. Figure 2b), while the CuSO_4_-treated pigs had a reduction in P-LCR (*p* = 0.036). The plasma Fe concentration in the Cu-Gly group was higher than that in the CuSO_4_ group (*p* = 0.002). No changes in the hematology parameters, including WBC, LYM, MID, GRAN, RBC, PLT, MCH, HCT, and PCT, were observed.

As compared to the baseline (d0), the supplementation of CuSO_4_ and Cu-Gly increased the plasma GH (*p* = 0.022, 0.018. Figure 2c) and IGF-1 levels (*p* = 0.030, 0.012. Figure 2d) at the end of the experiment. The Cu-Gly group had a higher plasma IGF-1 level (*p* = 0.039) than the CuSO_4_ group at the end of the experiment. Moreover, there was no difference in the plasma GH level between the CuSO_4_ and Cu-Gly groups. 

### 3.3. Fecal Microbiome

A total of 1280355 valid reads obtained from 24 samples were assigned to 1185 operational taxonomic units (OTUs) at the 97% similarity level. Rarefaction curves of the Sobs, Shannon, and Simpson indexes revealed a sufficient sequencing and sample size, and most diversity was captured. As shown in the Venn diagram (Figure 3a), the numbers of OTUs in the CuSO_4_ (1021 OTUs) and Cu-Gly groups (1053 OTUs) were reduced by 5.20% and 4.01%, respectively, as compared with the baseline (1077 OTUs for CuSO_4_ and 1097 OTUs for Cu-Gly).

The results of alpha diversity are presented in Figure 4. The treatments of Cu-Gly and CuSO_4_ had a decrease in the Shannon index (*p* < 0.001) and an increase in the Simpson index (*p* = 0.014, 0.005) of the fecal microbiome at the end of the experiment, as compared with the baseline, and decreases in Ace (*p* = 0.004) and Chao (*p* = 0.06) were observed in the CuSO_4_ treatment. There was no change in the Ace and Chao indexes with Cu-Gly supplementation during the experiment. At the end of the experiment, the Shannon index of the Cu-Gly group was higher (*p* < 0.001) and the Simpson index of the Cu-Gly group was lower (*p* = 0.035) than those of the CuSO_4_ group.

Furthermore, beta diversity revealed that the microbiota community was structurally changed by supplemental CuSO_4_ and Cu-Gly. The two-dimensional plot of principal component analysis (PCA) indicated that the CuSO_4_ group displayed a distinct separation pattern from the baseline and Cu-Gly groups (Figure 3b). The composition of fecal microbiota was analyzed, and *Firmicutes* was the predominant phylum, varying from 67.93% to 75.37%, followed by *Bacteroideteds* (12.18–17.68%) and *Spirochaetes* (10.42–12.08%). At the family level, the dominant floras were comprised of *Peptostreptococcaceae*, *Clostridiaceae*, *Lactobacillaceae*, *Ruminococcaceae*, *Spirochaetaceae*, *Prevotellaceae*, *Lachnosiraceae*, and *Christensenellaceae*, which represent more than 80% of the diversity. Compared with the baseline (d0), CuSO_4_ increased the relative percent of the family *Peptostreptococcaceae*, *Clostridiaceae*, and *Prevotellaceae* and decreased the percent of the family *Ruminococcaceae*, *Lactobacillaceae*, and *Lachnospiraceae* (Figure 3c). Interestingly, a contrary trend in the percent of community abundance was observed after the Cu-Gly treament.

At the end of the experiment, CuSO_4_ increased the relative abundance of *Synergistetes* phylum compared to Cu-Gly (*p* = 0.021), though the proportion is tiny (<0.005%), and it was absent in feces at d0 (Figure 5a). Compared with the baseline, Cu-Gly-treated pigs had an increase in the abundance of the *Lachnospiraceae* family (*p* = 0.043. Figure 5c) and *Lachnospiraceae_XPB1014_group* genus (*p* = 0.046. Figure 5d) and a decrease in the *Lachnospiraceae_NK4B4_group* genus (*p* = 0.043. Figure 5h), while the CuSO_4_-treated pigs had an increase in the abundance of the *Peptostreptococcaceae* family (*p* = 0.035. Figure 5b) and a decrease in the abundance of the family *Lachnospiraceae* (*p* = 0.024) and the genera *Corprococcus_3* (*p* = 0.047. Figure 5e), *Anaerorhabdus_furcosa_group* (*p* = 0.042. Figure 5f), *Lachnospiraceae_FCS020_group* (*p* = 0.008. Figure 5g), and *Lachnospiraceae_NK4B4_group* (*p* = 0.002). As compared with CuSO_4_, Cu-Gly had a higher abundance of the family *Lachnospiraceae* (*p* = 0.008) and the genera *Lachnospiraceae XPB1014* (*p* = 0.038), *Corprococcus_3* (*p* = 0.039), *Anaerorhabdus_furcosa_group* (*p* = 0.023), *Lachnospiraceae_FCS020_group* (*p* = 0.033), and *Lachnospiraceae_NK4B4_group* (*p* = 0.048) and a lower abundance of the *Synergistetes* phylum (*p* = 0.021) and the *Peptostreptococcaceae* family (*p* = 0.029) than CuSO_4_. 

The results of the linear discriminant analysis effect size (LEfSe) indicated that the family *Lachnospiraceae* and the genera *Lachnospiraceae* (*XPB1014*, *ND3007*, *NK4B4*, and *FCS020*), *Solobacterium*, *Coprococcus_3*, *Ruminococcus_2*, *Ruminococcus_gauvreauii_group*, *Fournierella*, *Eubacterium_eligens_group*, *Ruminococcus_gauvreauii_group*, *Faecalibacterium*, and *Prevotella_9* dominantly contributed to the significant difference in the microbial community of the Cu-Gly group (Figure 5i). However, the phylum *Synergistetes*, the class *Synergistaceae*, the order *Synergistales*, the family *Synergistaceae*, and the genera *Prevotellaceae_UCG-004* and *Cloacibacillus* dominantly contributed to the significant difference in the microbial community of the CuSO_4_ group.

### 3.4. Metabolomic Profiles

The LC-MS analysis detected a total of 775 peaks in all samples and observed 167 differential metabolites based on the variable importance in the projection (VIP) > 1 and *p* < 0.05. Furthermore, according to the accurate masses and MS spectrum measured by UHPLC, 22 metabolites in the metabolomics databases (HMDB) were identified based on the parameters of fold change (FC) > 1.30 or FC < 0.77 (Table 3). Differential metabolites between the CuSO_4_ and Cu-Gly groups were enriched in the Kyoto Encyclopedia of Genes and Genomes (KEGG) pathways of pyrimidine metabolism, ATP binding cassette (ABC) transporters, purine metabolism, protein digestion and absorption, and aminoacyl-tRNA biosynthesis (Figure 6).

After two weeks of the experiment, the fecal concentrations of 3-hydroxytridecanoic acid (*p* = 0.034, 0.009), N-acetylmuramate (*p* = 0.002, 0.003), 5-[(E)-2-(3,5-dihydroxyphenyl) ethenyl]-2-methoxyphenyl oxidanesulfonic acid (*p* = 0.005, 0.012), physagulin F (*p* = 0.007, 0.031), alpha-carboxy-delta-decalactone (*p* = 0.013, 0.014), and 5-pentyltetrahydro-2-oxo-3-furancarboxylic acid (*p* = 0.033, 0.026) increased in both the CuSO_4_ and Cu-Gly groups. As compared with the baseline, CuSO_4_ decreased the concentration of zizybeoside I (*p* = 0.017) and increased the concentrations of grevilline B (*p* = 0.014), 7-hydroxy-1-oxo-1H-isochromene-3-carbaldehyde (*p* = 0.043), 3-hydroxypristanic acid (*p* = 0.036), ricinoleic acid (*p* = 0.004), 3,4,5-trihydroxy-6-(2-hydroxy-1,2-diphenylethoxy) oxane-2-carboxylic acid (*p* = 0.013), 23-hydroxyphysalolactone (*p* = 0.015), pristanoylglycine (*p* = 0.002), trans-grandmarin (*p* = 0.016), and tyrosyl-Serine (*p* = 0.020). Cu-Gly decreased the concentration of 20-Oxo-leukotriene E4 (*p* = 0.019) and increased the concentrations of Nb-p-Coumaroyl-tryptamine (*p* = 0.031), coumarin (*p* = 0.004) and coutaric acid (*p* < 0.001), prolyl-arginine (*p* = 0.021), and Xi-Linalool 3-[rhamnosyl-(1->6)-glucoside] (*p* = 0.001) at the end of the experiment. As shown in Figure 7, the Cu-Gly group had an increase in the concentrations of propionic acid (*p* = 0.012), butyric acid (*p* = 0.005), pentanoic acid (*p* = 0.010), isopentanoic acid (*p* = 0.001), and total SCFAs (*p* = 0.030) compared with the baseline, while the CuSO_4_ group had an increase in the concentrations of acetic acid (*p* = 0.003) and isopentanoic acid (*p* = 0.041). 

Compared with the CuSO_4_ group, the Cu-Gly group had higher concentrations of butyric acid (*p* = 0.019), pentanoic acid (*p* = 0.029), isopentanoic acid (*p* = 0.008), Nb-p-Coumaroyl-tryptamine (*p* = 0.029), and coumarin (*p* = 0.024) and a lower concentration of 20-Oxo-leukotriene E4 (*p* = 0.003). No difference in the fecal concentrations of propionic acid and total SCFAs was observed between the Cu-Gly and CuSO_4_ groups.

### 3.5. Correlation Analysis

The correlation analysis showed that the *Synergistetes* phylum was positively correlated with the fecal Cu content (r = 0.39, *p* = 0.050. Figure 8a) and negatively correlated with the concentration of fecal butyric acid (r = −0.68, *p* = 0.030. Figure 8b). The *Proteobacteria* phylum was positively correlated with the plasma Fe concentration (r = 0.64, *p* = 0.047) and negatively correlated with the fecal Cu (r = −0.38, *p* = 0.046) and Mn contents (r = −0.18, *p* = 0.033). The *Peptostreptococcaceae* family was negatively correlated with the concentrations of plasma Fe (r = −0.62, *p* = 0.045), fecal propionic acid (r = −0.62, *p* = 0.049), and pentanoic acid (r = −0.75, *p* = 0.012). The *Lachnospiraceae* family was negatively correlated with the fecal Cu content (r = −0.63, *p* = 0.049) and positively correlated with the concentrations of fecal methyl dioxindole-3-acetate (r = 0.60, *p* = 0.039. Figure 8c), methyl 1-methoxy-1H-indole-3-carboxylate (r = 0.70, *p* = 0.012), butyric acid (r = 0.70, *p* = 0.024), plasma GH (r = 0.43, *p* = 0.034), and Fe (r = 0.63, *p* = 0.049). The genus *Lachnospiraceae_XPB1014_group* was positively correlated with the concentrations of fecal coumarin (r = 0.64, *p* = 0.025), methyl dioxindole-3-acetate (r = 0.76, *p* = 0.004), methyl 1-methoxy-1H-indole-3-carboxylate (r = 0.84, *p* = 0.001), gamma-glutamylphenyalanine (r = 0.59, *p* = 0.045), butyric acid (r = 0.77, *p* = 0.009), and plasma Fe (r = 0.66, *p* = 0.037). The genus *Lachnospiraceae_NK4A136_group* was negatively correlated with the contents of fecal Zn (r = −0.19, *p* = 0.038) and Fe (r = −0.24, *p* = 0.048). The *Terrisporobacter* genus (*Peptostreptococcaceae* family) was negatively correlated with the concentrations of propionic acid (r = −0.85, *p* = 0.002) and pentanoic acid (r = −0.64, *p* = 0.044). There was no significant correlation between microbiota and other parameters, including IGF-1, HGB, MPV, PDW, and P-LCR.

## 4. Discussion

Organic Cu chelated with amino acids or peptides has been reported to improve the growth performance of pigs [4,5,6], with more retention [21] and less Cu excretion [22]. The absorption of organic Cu is different from that of inorganic Cu, though the mechanisms remain controversial. Organic Cu could be absorbed as compounds and hydrolyzed on the brush border, which alleviates the antagonisms and contributes to the improvement in mineral absorption [15]. Consistently, our data showed that pigs fed with Cu-Gly had a higher feed intake and microelement absorption and lower excretion. The GH-IGF-1 axis plays a critical role in the regulation of animal growth, and it is a target for the nutritional regulation of animal growth [23]. IGF-1 secretion is stimulated by GH and is mainly produced in the liver [24], which is a key tissue for Cu metabolism and storage. It was suggested that the serum GH and IGF-1 concentrations of piglets could be elevated by 160 mg/kg Cu as Cu methionine hydroxy analog chelated [7] or by 100–300 mg/kg Cu as CuSO_4_ [25]. There was no difference in the serum GH concentration between CuSO_4_ and Cu methionine (Cu-Met) in growing pigs [26]. A previous study suggested that the decreased feed intake and plasma concentrations of GH and IGF-1 were accompanied by a reduced abundance of the *Lachnospiraceae* family in the caecum of weaned piglets [27]. Consistently in this study, Cu-Gly had a greater stimulation regarding the secretion of IGF-1, but not GH, than CuSO_4_. Cu-Gly supplementation increased the abundance of the *Lachnospiraceae* family, which was negatively correlated with the fecal Cu content and positively correlated with the plasma Fe and GH concentrations. The intestinal Fe absorption was closely correlated with *Lachnospiraceae*, and the intestinal absorption of minerals depended on the concentration of butyrate [28]. This indicated that the Cu-Gly treatment elevated plasma IGF-1 and improved the absorption of Cu, Fe, and Mn compared with the CuSO_4_ treatment, which might be attributed to the elevated *Lachnospiraceae* family in the hindgut.

Trace elements, such as Cu, Zn, and Fe, have an influence on the intestinal microenvironment, including the microbiota and nutrient digestibility [18]. The antimicrobial properties of Cu at high doses that have been documented though the underlying mechanism have yet to be elucidated [29,30]. The levels of dietary Fe can modulate the gut microbiome and microbial metabolites, especially for the butyrate content in the colon [31]. On the other hand, gut microbiota also play a role in the absorption and metabolism of trace elements, and there is a complicated mechanism modulating the mineral absorption in the gut [14]. Probiotics such as *Lactobacilli* can improve Fe absorption by facilitating the phytate degradation and iron release from diets [32]. Phytic acid prevents the absorption of essential trace elements by binding the metallic ions, and phytase inclusion is recommended in swine diets to improve the digestibility and utilization of trace elements, as well as calcium and phosphorus [33]. Because of using wheat bran at a high level in the present study, phytic acid at a high level (0.51% to 0.66%) in diets could promote the interaction with trace elements in the digesta and have a great influence on the mineral digestibility and metabolism [34,35]. Phytase was added to the basal diets as a recommendation, and the same conventional dose (500 U/kg diet) was used in both the basal and experimental diets in order to exclude the interference. Consistently, our results suggested that the Cu-Gly supplementation increased the content of fecal butyric acid and improved the absorption of Cu, Fe, and Mn as compared with the CuSO_4_ supplementation. Meanwhile, Cu-Gly increased the abundance of the *Lachnospiraceae* family and *Lachnospiraceae_XPB1014_group* genus, while the CuSO_4_ supplementation increased the abundance of the *Peptostreptococcaceae* family and decreased the abundance of the *Lachnospiraceae* family and the genera *Corprococcus_3*, *Anaerorhabdus_furcosa_group*, *Lachnospiraceae_FCS020_group*, and *Lachnospiraceae_NK4B4_group*. Furthermore, the precise interactions among the gut microbiome and mineral absorption and the impact of butyric acid in the modulation of the Cu source on the gut microbiota and health need further investigations, which will be helpful for us to understand it better.

The *Peptostreptococcaceae* family was elevated in ulcerative colitis patients, which is closely related to intestinal inflammation [36]. A study in cows indicated that the improvement in nutrient digestibility was associated with a decreased *Peptostreptococcaceae* abundance and increased plasma propionate concentration [37]. Both *Synergistetes* and *Lachnospiraceae* are identified as SCFA-producing bacteria [38]. The main products of *Synergistetes* are acetate, propionate, and formate [39], while the main product of *Lachnospiraceae* is butyrate [40]. It was reported that *Synergistetes* could reduce plant toxins by degrading fluoroacetate into acetate [41]. *Synergistetes* also participated in the anaerobic dissimilation of acetate into methane and carbon dioxide [42], which leads to a loss in the gross energy intake and is associated with a lower energy efficiency [43]. The main products of the genera *Lachnospiraceae_XPB1014_group* and *Coprococcus_3* are propionate and butyrate, which play a beneficial role in the lipid metabolism [44] and anti-inflammatory response [45]. Butyrate is known as a primary energy source for enterocytes and also provides several benefits to intestinal immunity and host health [46]. The gut microbiome could modulate the host immunity and health by altering the butyrate producers [47]. Intestinal inflammation could be alleviated by butyrate administration or elevating butyrate-producing bacteria [48]. A previous study suggested that pigs of a high feed efficiency had a higher abundance of *Lachnospiraceae* [49]. Consistently, here, the fecal concentration of butyric acid was positively correlated with the abundance of the family *Lachnospiraceae* and the genus *Lachnospiraceae_XPB1014_group* and negatively correlated with the abundance of the phylum *Synergistetes*. Meanwhile, the fecal concentration of propionic acid was negatively correlated with the abundance of the *Peptostreptococcaceae* family and the *Terrisporobacter* genus (belongs to the *Peptostreptococcaceae* family). Compared with the CuSO_4_ group, the Cu-Gly group had a higher abundance of the *Lachnospiraceae* family and the genera *Lachnospiraceae_XPB1014_group* and *Corprococcus_3* (belongs to the *Lachnospiraceae* family) and a lower abundance of the *Synergistetes* phylum and the *Peptostreptococcaceae* family. These findings supported the idea that Cu-Gly and CuSO_4_ altered the gut microbial communities and SCFAs production in a different manner, by which they had an influence on the intestinal health.

Furthermore, pigs supplemented with Cu-Gly had higher concentrations of Nb-p-Coumaroyl-tryptamine and coumarin and a lower concentration of 20-oxo-leukotriene E4 compared with those supplemented with CuSO_4_ in the present study. Nb-p-Coumaroyl-tryptamine and its derivatives have excellent anti-oxidant and anti-inflammatory activities [50]. Nb-p-Coumaroyl-tryptamine suppressed the LPS-induced extracellular secretion of pro-inflammatory cytokines in macrophage cells [51]. Leukotriene E4 is an inflammation mediator involved in the activation of the inflammatory process [52]. The 20-oxo-leukotriene E4 is a metabolite in the lipid oxidation of leukotriene E4, which is a potential metabolic marker of inflammation [53]. These findings supported the idea that Cu-Gly and CuSO_4_ differentially affected metabolic profiles in the hindgut, which partially contributed to the different effects of the Cu source on the host health. The dietary composition could greatly affect the microbial communities, and metabolites produced from dietary components or biochemically modified by gut bacteria [54] mediate the interaction between the microbes and host [55,56]. Coumarins are the phenolic compounds with antioxidant and anti-inflammation activities [57], which have been used to improve intestinal oxidative status in the treating gut inflammation [58]. Coumarins and coumarin derivatives could ameliorate inflammatory bowel diseases by inhibiting lipid peroxidation [59], which was accompanied by a high abundance of *Lachnospiraceae* [60]. Consistently, the integrated microbiome and metabolome analysis suggested that *Lachnospiraceae_XPB1014_group* was elevated by Cu-Gly and positively correlated with the concentrations of fecal coumarin and butyric acid. Our data presented the correlations between fecal microbiota and the metabolomics response and fecal or plasma indicators, which provide us with new insights into the potential interactions. Due to the limitation of the small sample size in the metabolic experiment, further investigations into the effects of Cu sources on the growth performance, gut microbiome, and immunity using a larger number of animals should be conducted. These findings highlight the microbiome-metabolome response to dietary Cu sources in growing pigs, and the potential benefits of the genus *Lachnospiraceae_XPB1014_group*, fecal coumarin, and butyric acid for intestinal health should be investigated in the future to understand it better.

## 5. Conclusions

Dietary Cu-Gly supplementation increased the feed intake and plasma IGF-1 and decreased fecal microelement contents as compared to CuSO_4_ supplementation. Cu-Gly supplementation elevated the abundance of the *Lachnospiraceae* family and the *Lachnospiraceae_XPB1014_group* genus, and CuSO_4_ supplementation increased the abundance of the *Synergistetes* phylum and the *Peptostreptococcaceae* family. Cu-Gly and CuSO_4_ altered microbial communities and fecal metabolites in different manners, which partially contributed to the differential effects on the intestinal health of pigs.

## Figures and Tables

**Figure 1 animals-13-00345-f001:**
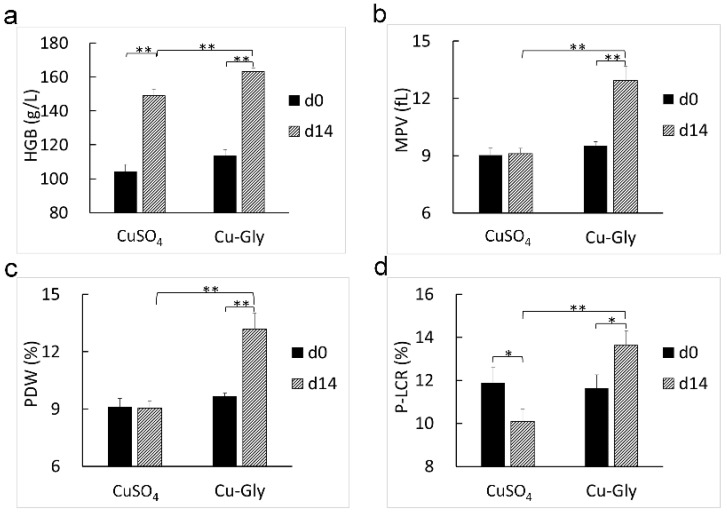
Effects of Cu sources on the hematology parameters of pigs. Hematology analysis was performed, and only a significant difference in hematology parameters, including HGB (**a**), MPV (**b**), PDW (**c**), and P-LCR (**d**), was shown. Data present mean values ± standard errors. * represents a significant difference at *p* < 0.05, and ** represents a significant difference at *p* < 0.01. n = 6. HGB, hemoglobin; MPV, mean platelet volume; PDW, platelet distribution width; P–LCR, platelet–larger cell ratio.

**Figure 2 animals-13-00345-f002:**
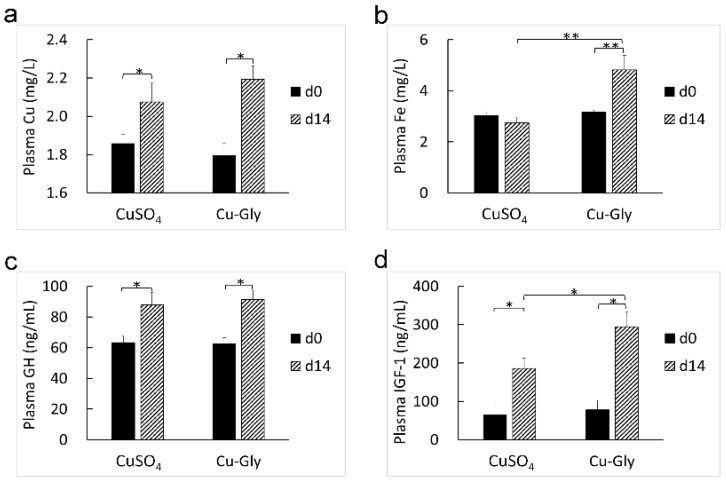
Effects of Cu sources on the levels of plasma Cu, Fe, GH, and IGF-1. Concentrations of plasma Cu (**a**), Fe (**b**), GH (**c**), and IGF-1 (**d**). Data present mean values ± standard errors. * represents a significant difference at *p* < 0.05, and ** represents a significant difference at *p* < 0.01. n = 6. GH, growth hormone; IGF-1, insulin like growth factor 1.

**Figure 3 animals-13-00345-f003:**
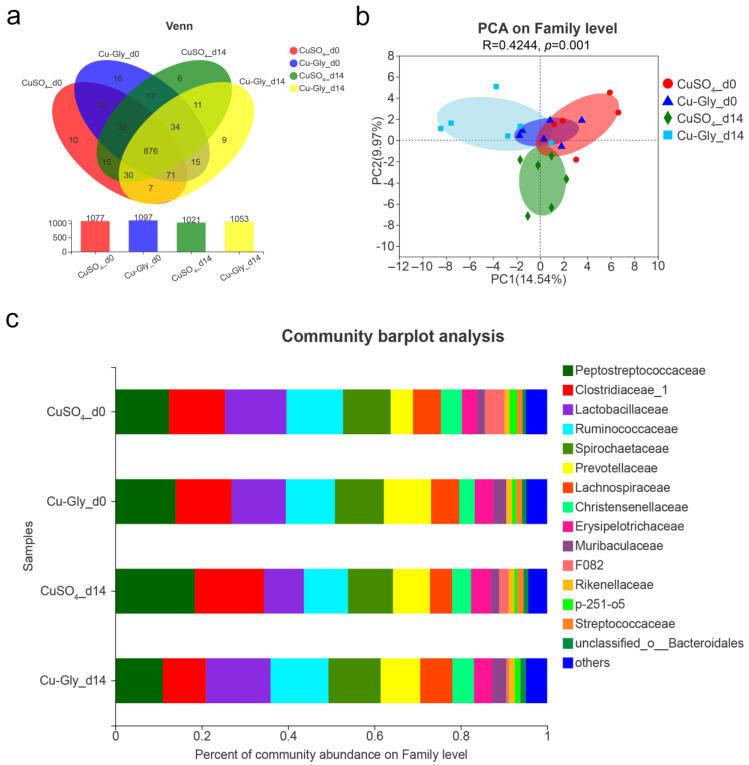
Changes in microbial communities in the feces of pigs. (**a**) Venn analysis at the OTU level; (**b**) Principal component analysis (PCA) score plots at the family level; (**c**) Taxonomic classification of the 16S rRNA gene sequences at the family level. n = 6.

**Figure 4 animals-13-00345-f004:**
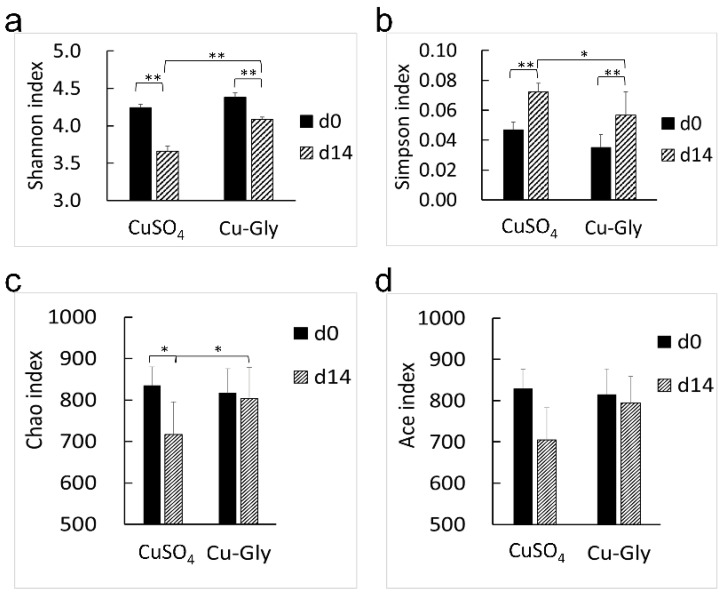
Effects of Cu sources on the alpha diversity of the microbial community in the feces of pigs. The Shannon (**a**), Simpson (**b**), Ace (**c**), and Chao (**d**) indices were calculated. Data present mean values ± standard errors. * represents a significant difference at *p* < 0.05, and ** represents a significant difference at *p* < 0.01. n = 6.

**Figure 5 animals-13-00345-f005:**
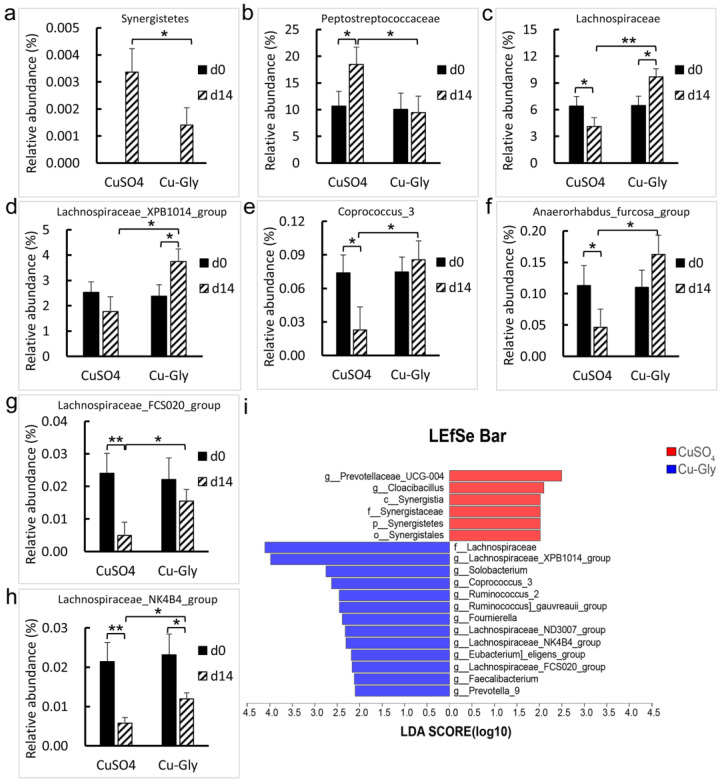
Effects of Cu sources on the relative abundance of bacteria in the feces of pigs. The column charts showed the changes in the relative abundance of the phylum *Synergistetes* (**a**), the families *Peptostreptococcaceae* (**b**) and *Lachnosiraceae* (**c**), and the genera *Lachnospiraceae_XPB1014_group* (**d**), *Corprococcus_3* (**e**), *Anaerorhabdus_furcosa_group* (**f**), *Lachnospiraceae_FCS020_group* (**g**), and *Lachnospiraceae_NK4B4_group* (**h**) in the CuSO_4_ and Cu-Gly groups. Only significant changes in microbiota abundance were presented. Data present mean values ± standard errors. * represents a significant difference at *p* < 0.05, and ** represents a significant difference at *p* < 0.01. (**i**) A non-parametric factorial Kruskal–Wallis (KW) sum-rank test and linear discriminant analysis (LDA) were performed to discriminate the microbial community that contributed to the significant difference in the CuSO_4_ and Cu-Gly groups. The bar represents thelLinear discriminant analysis effect size (LEfSe). n = 6.

**Figure 6 animals-13-00345-f006:**
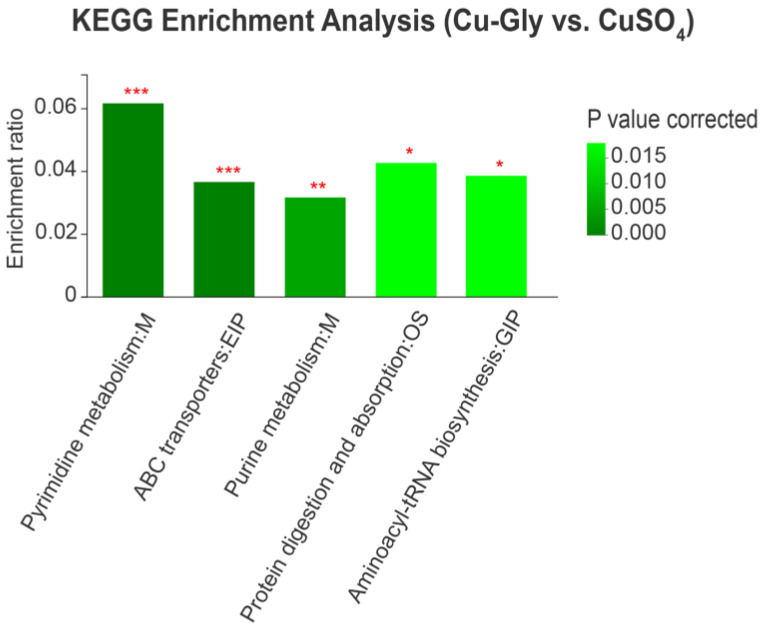
Differential metabolites were annotated, categorized, and enriched to the KEGG pathways. * represents a significant difference at *p* < 0.05, ** represents a significant difference at *p* < 0.01, *** represents a significant difference at *p* < 0.001. n = 6. KEGG, Kyoto Encyclopedia of Genes and Genomes.

**Figure 7 animals-13-00345-f007:**
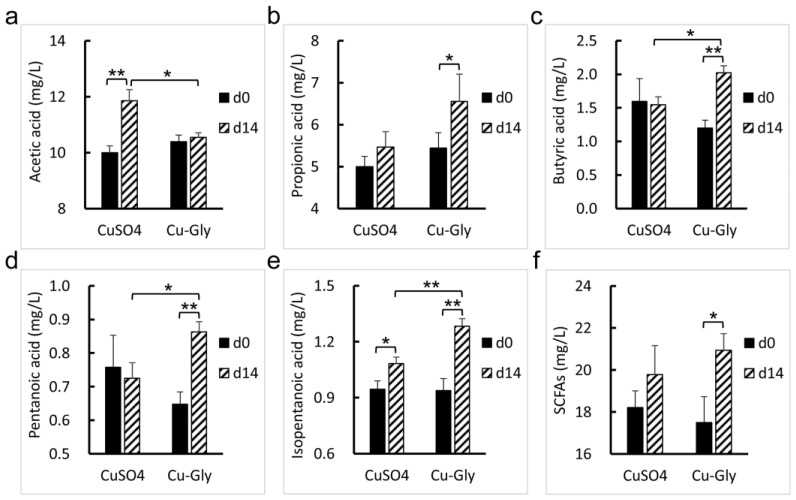
Effects of Cu sources on the concentrations of SCFAs in the feces of pigs. The concentrations of acetic acid (**a**), propionic acid (**b**), butyric acid (**c**), pentanoic acid (**d**), isopentanoic acid (**e**), and SCFAs (**f**) were determined. Data present mean values ± standard errors. * represents a significant difference at *p* < 0.05, and ** represents a significant difference at *p* < 0.01. n = 6. SCFAs, short-chain fatty acids.

**Figure 8 animals-13-00345-f008:**
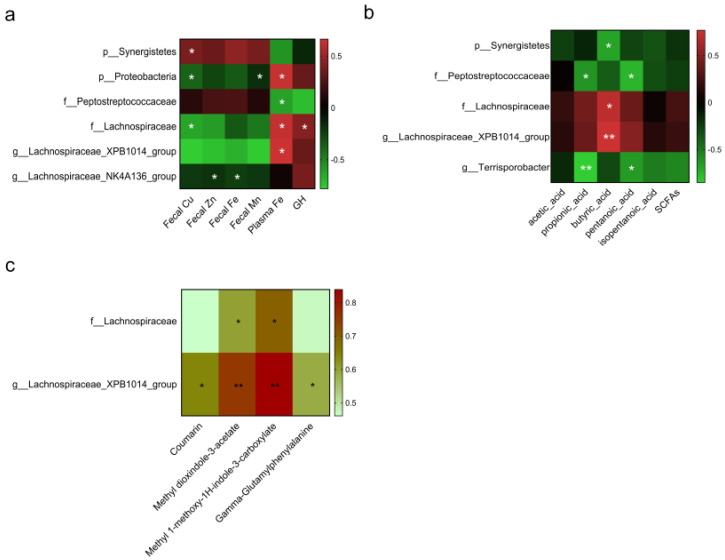
Correlation analysis between microbiota and metabolites, SCFAs, or fecal and plasma indicators. Spearman correlation analysis between differential microbiota and plasma indicators, fecal trace elements (**a**), SCFAs (**b**), and metabolites (**c**). Only significant correlations were presented. * represents a significant difference at *p* < 0.05, and ** represents a significant difference at *p* < 0.01. n = 6. GH, growth hormone; SCFAs, short-chain fatty acids.

**Table 1 animals-13-00345-t001:** Ingredients and composition of the basal diet ^1^.

Ingredient	Content (%)	Nutrient Composition	
Corn	65.30	Metabolic energy (MJ/kg)	12.21
Soybean meal	18.40	Crude protein ^4^ (%)	16.21
Wheat bran	12.20	Calcium (%)	0.71
Limestone	1.12	Total phosphorus (%)	0.55
Dicalcium phosphate	0.85	Lysine (%)	1.12
Salt	0.30	Methionine + cysteine (%)	0.65
Phytase ^2^	0.01	Threonine (%)	0.72
L-Lysine hydrochloride	0.50	Tryptophan (%)	0.18
DL-Methionine	0.16		
Threonine	0.16		
Premix ^3^	1.00		

^1^ The intrinsic copper level of the basal diet is 5.7 mg/kg. ^2^ The calculated level of phytic acid varies from 0.51% to 0.66%, and the premix provided 500 U phytase per kilogram of diet. ^3^ The premix provided per kilogram of diet: 60 mg Zn, 150 mg Fe, 85 mg Mn, 0.3 mg Se and 0.14 mg I, 2000 IU vitamin A, 1500 IU Vitamin D3, 53 mg vitamin E, 1 mg vitamin K3, 6 mg thiamine, 2.8 mg riboflavin, 2.8 mg vitamin B6, 0.1 mg vitamin B12, 2 mg folic acid, 8 mg niacin, 28 mg calcium pantothenate, and 0.2 mg biotin. ^4^ Determined by chemical analysis.

**Table 2 animals-13-00345-t002:** Effects of Cu sources on growth performance, digestibility, and fecal microelement contents.

Item	CuSO_4_	Cu-Gly	SEM	*p*-Value
Initial BW (kg)	24.78	24.92	2.96	0.707
Final BW (kg)	31.86	32.72	3.36	0.509
ADG (g/d)	505.43	557.43	57.08	0.227
ADFI (g/d)	1348.29	1483.57	101.07	0.039
Feed efficiency (feed/gain)	2.67	2.66	0.629	0.258
ATTD ^1^ of DM (%)	90.44	90.45	0.25	0.949
ATTD of CP (%)	83.91	84.72	1.75	0.656
ATTD of Cu (%)	45.17	56.49	5.03	0.048
ATTD of Zn (%)	26.11	27.55	4.01	0.727
ATTD of Fe (%)	52.62	67.49	5.38	0.021
ATTD of Mn (%)	49.46	61.57	5.41	0.048
Fecal content of DM (%)	25.37	26.18	1.09	0.475
Fecal content of CP (%)	47.70	48.09	1.11	0.730
Fecal content of Cu (mg/kg)	522.04	484.28	14.04	0.023
Fecal content of Zn (mg/kg)	1062.75	979.68	65.50	0.270
Fecal content of Fe (mg/kg)	1805.68	1120.97	230.41	0.016
Fecal content of Mn (mg/kg)	1047.37	885.47	67.47	0.040

^1^ ATTD, apparent total tract digestibility. n = 6. BW, body weight; ADG, average daily gain; ADFI, average daily feed intake; DM, dry matter; CP, crude protein.

**Table 3 animals-13-00345-t003:** Effects of Cu sources on the metabolite concentrations in the feces of pigs.

Metabolite	HMDB ID	CuSO_4_ vs. Baseline		Cu-Gly vs. Baseline		Cu-Gly vs. CuSO_4_	
		FC ^1^	*p*-Value	FC	*p*-Value	FC	*p*-Value
3-hydroxytridecanoic acid	HMDB0061655	3.697	0.034	3.528	0.009	-	-
N-Acetylmuramate	HMDB0060493	2.188	0.002	2.075	0.003	-	-
5-[(E)-2-(3,5-dihydroxyphenyl)ethenyl]-2-methoxyphenyl oxidanesulfonic acid	HMDB0134587	1.699	0.005	1.593	0.012	-	-
Physagulin F	HMDB0039631	1.527	0.007	1.418	0.031	-	-
Grevilline B	HMDB0033240	1.498	0.014	-	-	-	-
7-hydroxy-1-oxo-1H-isochromene-3-carbaldehyde	HMDB0128617	1.456	0.043	-	-	-	-
3-hydroxypristanic acid	HMDB0061651	1.417	0.036	-	-	-	-
Ricinoleic acid	HMDB0034297	1.390	0.004	-	-	-	-
Alpha-Carboxy-delta-decalactone	HMDB0030985	1.388	0.013	1.410	0.014	-	-
3,4,5-trihydroxy-6-(2-hydroxy-1,2-diphenylethoxy)oxane-2-carboxylic acid	HMDB0135201	1.371	0.013	-	-	-	-
5-Pentyltetrahydro-2-oxo-3-furancarboxylic acid	HMDB0030989	1.369	0.033	1.413	0.026	-	-
23-Hydroxyphysalolactone	HMDB0031388	1.355	0.015	-	-	-	-
Pristanoylglycine	HMDB0013303	1.350	0.002	-	-	-	-
Trans-Grandmarin	HMDB0039030	1.347	0.016	-	-	-	-
Tyrosyl-Serine	HMDB0029114	1.326	0.020	-	-	-	-
Zizybeoside I	HMDB0034954	0.752	0.017	-	-	-	-
Prolyl-Arginine	HMDB0029011	-	-	2.052	0.021	-	-
Xi-Linalool 3-[rhamnosyl-(1->6)-glucoside]	HMDB0030422	-	-	1.360	0.001	-	-
Coutaric acid	HMDB0029225	-	-	1.343	0.000	-	-
Nb-p-Coumaroyltryptamine (CT)	HMDB0041518	-	-	1.474	0.031	1.388	0.036
Coumarin	HMDB0001218	-	-	1.421	0.004	1.440	0.024
20-Oxo-leukotriene E4	HMDB0012642	-	-	0.713	0.019	0.757	0.003

^1^ FC, fold change in the mean value of the peak area obtained from each group. - indicates no statistical significance at *p* > 0.05 or 0.77 < FC < 1.3. n = 6. FC, fold change; HMDB, human metabolome database.

## Data Availability

The data supporting the conclusions of this article are all available.

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
