# Peer review of "Comparison of the Microbiome-Metabolome Response to Copper Sulfate and Copper Glycinate in Growing Pigs"

_animals, 2023, doi:10.3390/ani13030345_

Round 1
Reviewer 1 Report
The manuscript compared the fecal microbiome-metabolome response to copper sulfate (CuSO4) and copper glycinate (Cu-Gly) in pigs. These results indicated that Cu-Gly and CuSO4 could differentially affect fecal microbiota and metabolites, which partially contributes to intestinal health of pigs in different manners. Overall, the research is interesting and meritorious. And the paper was generally well-written and –structured. Nonetheless, these are still some issues should be addressed.
1. I suggested the author should adjust the order of Figure 6 and Figure 7, because the Figure 6 is in the latter chapter in the text.
2. In the “Correlation Analysis” part, the spearman correlation need to be interpreted with caution due to the small sample size (n =6). Therefore, I suggested adding some statement about to the limitation of this study in the discussion part, such as the sample size.
3. In line 326-329, the author wrote “Furthermore, according to the accurate masses and MS spectrum measured by UHPLC, 22 metabolites in the metabolomics databases (HMDB) were identified as the potential markers for Cu supplementation based on the parameters of fold change (FC) > 1.30 or FC < 0.77 (Table 3).” It is not suitable to define the “potential markers” according to the down-regulated metabolites in Cu supplementation group.
Reviewer 2 Report
The manuscript is interesting and contains results that deserve to be published. There are minor suggestions from my side.
Intoduction.- please provide an hypothesis which justify the experimental design and studied variables
Diet contain a high level of wheat bran and then a high level of phytic acid can be calculated. Please, provide information about the phytase activity supplementation in the diets. A high level of phytic acid could promote interaction with trace minerals in the digesta, which could explain some of the observed changes.
Was P and Ca digestibility registered? if phytic acid or phytase effective activity is involved on the results, P digestibility could help to explore alternative mechanisms.
The manuscript provide interesting results about the effect of different trace mineral sources on the digesta microbiota and metabolomic results of growing pigs when included to low levels in the diet. Results evidence significant effects and correlations between microbiota and different metabolites which have been properly discussed. However, the discussion section lacks of discussion to explain why or how different Cu sources are able to modify the mineral absorption or interact with the intestinal microbiota. Is a direct effect of trace minerals availability? is result of an interaction among trace minerals and phytic acid? and likely interactions with the phytase efficacy? If Cu source is in the origin, and microbiota may mediates the response, it is necessary to try to understand how evolve the events in the digestive tract.
